# Role of AMPK in Regulation of Oxaliplatin-Resistant Human Colorectal Cancer

**DOI:** 10.3390/biomedicines10112690

**Published:** 2022-10-25

**Authors:** Sun Young Park, Ye Seo Chung, So Yeon Park, So Hee Kim

**Affiliations:** 1College of Pharmacy and Research Institute of Pharmaceutical Science and Technology, Ajou University, Suwon 16499, Korea; 2Department of Biohealth Regulatory Science, Graduate School of Ajou University, Suwon 16499, Korea

**Keywords:** colorectal cancer, oxaliplatin, chemoresistance, Akt-mTOR, AMPK, autophagy, glycolysis

## Abstract

Oxaliplatin is a platinum analog that can interfere with DNA replication and transcription. Continuous exposure to oxaliplatin results in chemoresistance; however, this mechanism is not well known. In this study, oxaliplatin-resistant (OR) colorectal cancer (CRC) cells of HCT116, HT29, SW480 and SW620 were established by gradually increasing the drug concentration to 2.5 μM. The inhibitory concentrations of cell growth by 50% (IC_50_) of oxaliplatin were 4.40–12.7-fold significantly higher in OR CRC cells as compared to their respective parental (PT) CRC cells. Phospho-Akt and phospho-mammalian target of rapamycin (mTOR) decreased in PT CRC cells but was overexpressed in OR CRC cells in response to oxaliplatin. In addition, an oxaliplatin-mediated decrease in phospho-AMP-activated protein kinase (AMPK) in PT CRC cells induced autophagy. Contrastingly, an increased phospho-AMPK in OR CRC cells was accompanied by a decrease in LC3B, further inducing the activity of glycolytic enzymes, such as glucose transporter 1 (GLUT1), 6-phosphofructo-2-kinase/fructose-2,6-bisphosphatase 3 (PFKFB3) and phosphofructokinase 1 (PFK1), to mediate cell survival. Inhibition of AMPK in OR CRC cells induced autophagy through inactivation of Akt/mTOR pathway and a decrease in GLUT1, PFKFB3, and PFK1. Collectively, targeting AMPK may provide solutions to overcome chemoresistance in OR CRC cells and restore chemosensitivity to anticancer drugs.

## 1. Introduction

According to 2020 statistics, colorectal cancer (CRC) accounts for 10% of all cancer incidences and 9.4% of mortality worldwide and is common in both men and women [1]. As with other cancer types, the cause of CRC remains unclear. Based on several studies, multiple stages and numerous factors are known to be involved in the pathogenesis of CRC [2]. The primary treatments for CRC include surgery, radiation therapy and chemotherapy [1,3]. However, repeated administration of chemotherapeutics has been associated with drug resistance and poor prognosis. Therefore, various attempts, such as targeted molecular therapy have been recently made to overcome drug resistance in CRC patients [4].

The current paradigm states that combination therapy should be the best treatment option because it should prevent the development of drug resistance and be more effective than any one drug on its own [5]. One of the mechanisms of chemoresistance is the increased expression of P-glycoprotein (P-gp), a drug efflux pump [6]. Overexpression of P-gp lowers the cellular concentration of the anticancer drugs, reduces their effects and induces chemoresistance [7]. Another mechanism of chemoresistance in cancer cells is the activation of the Akt-mammalian target of the rapamycin (mTOR) signaling pathway [8,9,10]. Akt regulates growth factors responsible for cell survival and delivers signals to mTOR. Activated mTOR plays a pivotal role in cell growth and proliferation [11,12,13]. Inhibition of activated Akt and its downstream effector, mTOR has been reported to induce autophagy, a form of apoptosis [14]. Autophagy is a survival system that maintains cellular homeostasis by degrading aged proteins and damaged intracellular organs and is thus involved in diseases and biological progression [15,16,17]. Autophagy is also regulated by AMP-activated protein kinase (AMPK) [18], which is evident through its induction in response to 5-aminoimidazole-4-carboxamide ribonucleotide or metformin, an AMPK activator and inhibition by compound C, an AMPK inhibitor [19]. The regulators of autophagy may, therefore, be attractive targets for CRC treatment. However, it is unclear whether the responsiveness of resistant cancer cells to anticancer drugs can be increased by regulating autophagy through the AMPK and Akt/mTOR pathways [20]. Furthermore, various investigators have suggested that fast-growing or resistant cancer cells have increased glycolysis for rapid energy supply [21]. Overexpression of hexokinase 2 (HK2) and glucose transporter 1 (GLUT1) is associated with chemoresistance in cancer [22,23,24].

Oxaliplatin, a platinum agent used for the treatment of CRC, forms a DNA-intra-strand crosslink that causes DNA deformation and eventually inhibits the synthesis of DNA and proteins in the cells, resulting in cytotoxic effects [25,26,27]. Oxaliplatin has been widely used for its anticancer effects against CRC and ovarian cancer, but its effectiveness is limited owing to the development of drug resistance [28,29,30].

In this study, we evaluated the effects of oxaliplatin resistance on autophagy in CRC. Changes in AMPK and Akt/mTOR signaling pathways, as a mechanism of oxaliplatin resistance, were examined in oxaliplatin-resistant (OR) CRC cells, extending the investigation of their effects on enzymes involved in glycolysis during the acquisition of oxaliplatin resistance in CRC cells.

## 2. Materials and Methods

### 2.1. Chemicals

Oxaliplatin, doxorubicin (DOX), 5-fluorouracil (5-FU), trichostatin A (TSA), 3-(4,5-dimethylthiazol-2-yl)-2,5-diphenyltetrazolium-bromide (MTT), dimethyl sulfoxide (DMSO), 4,6-diamidino-2-phenylindole (DAPI), metformin, compound C, cell lysis buffer and protease inhibitor cocktail were purchased from Sigma-Aldrich (St. Louis, MO, USA). Fetal bovine serum (FBS), Dulbecco’s modified Eagle’s medium (DMEM), Dulbecco’s phosphate-buffered saline (DPBS) and penicillin–streptomycin were purchased from Invitrogen (Carlsbad, CA, USA). Antibodies against breast cancer resistance protein (BCRP), multidrug resistance protein 2 (MRP2), p53 and glyceraldehyde 3-phosphate dehydrogenase (GAPDH) were obtained from Santa Cruz Biotechnology (Santa Cruz, CA, USA). Primary monoclonal antibodies against P-gp, Bax, cyclin D1, cyclin-dependent kinase 4 (CDK4), Bcl-_xL_, poly ADP ribose polymerase (PARP), Bad, phospho-Akt (Ser473), phospho-mTOR (Ser2448), phospho-S6RP, phospho-p70S6K (Ser424/Thr421), phospho-p70S6K (Thr389), AMPK, phospho-AMPK (Thr172), GLUT1, HK2, phosphofructokinase 1 (PFK1), pyruvate kinase M2 (PKM2), lactate dehydrogenase A (LDHA), 6-phosphofructo-2-kinase/fructose-2,6-biphosphatase 3 (PFKFB3) and LC3B were purchased from Cell Signaling Biotechnology (Danvers, MA, USA). Propidium iodide and RNase A were obtained from Abcam (Cambridge, UK). All other chemicals and reagents were of analytical or HPLC grade and were used as received without further purification.

### 2.2. Cell Culture and Establishment of Oxaliplatin Resistance in CRC Cells

Human CRC cell lines, HCT116, HT29 SW480 and SW620, were obtained from the Korean Cell Line Bank (KCLB, Seoul, Korea) and cultured in DMEM supplemented with 10% FBS and 1% antibiotics (100 U/mL penicillin/10 mg/mL streptomycin). The cells were incubated overnight in a humidified cell incubator with 5% CO_2_ at 37 °C allowed to attach.

OR CRC cells (HCT116/OR, HT29/OR, SW480/OR and SW620/OR) were established according to a previously reported method [31]. A stock solution of oxaliplatin was prepared by dissolving it in DPBS. Parental (PT) CRC cells (HCT116/PT, HT29/PT, SW480/PT and SW620/PT) were cultured in serum-containing media supplemented with 0.2 μM oxaliplatin. While this initial oxaliplatin treatment induced the death of most cells, some cells survived. However, these remaining cells proliferated slower than their respective PT CRC cells. At an 80% confluency, the cells were re-cultured in fresh culture medium supplemented with the same concentration of oxaliplatin. By continuously growing cells in the presence of the same concentration of oxaliplatin until no more detectable death of cells was observed, the concentration of oxaliplatin was increased by 0.2 μM. This step was repeated several times with a stepwise increase of 0.2 μM in oxaliplatin concentration to a maximum of 2.5 μM over approximately 5–6 months, OR CRC cells were established. For comparison, single treatment (ST) of oxaliplatin at 2.5 μM was also applied to the PT CRC cells (HCT116/ST, HT29/ST, SW480/ST and SW620/ST) for 24 h.

### 2.3. Cell Proliferation Assay

To determine the effect of oxaliplatin resistance on the proliferation of CRC cells, HCT116/PT, HT29/PT, SW480/PT and SW620/PT CRC cells and their respective OR CRC cells were seeded on 96-well plates. The cells were treated with appropriate concentrations of oxaliplatin, DOX, 5-FU and TSA. After 72 h of incubation, the medium was removed, diluted MTT solution (100 μL) was added, and the plate was incubated at 37 °C for an additional 2 h. Subsequently, the purple formazan crystals were solubilized by adding DMSO, the absorbance was read at 540 nm using an ELISA reader (Bio-Tek Instruments Inc., Winooski, VT, USA), and the inhibitory drug concentrations of cell growth by 50% (IC_50_) were estimated using GraphPad Prism 5 software (La Jolla, CA, USA).

To investigate the effects of AMPK on cell proliferation in ST and OR CRC cells, ST and OR CRC cells were treated with 10 mM metformin or 10 μM compound C for 72 h. The remaining procedures were the same as described above.

### 2.4. Flow Cytometry

To assess cell cycle progression, PT CRC cells and their respective OR CRC cells were plated on 60 mm dishes. After overnight incubation, the PT and OR CRC cells were treated with DPBS or oxaliplatin (2.5 μM) for 24 h [32,33]. Cells were then trypsinized and centrifuged for 5 min at 500× *g* at 4 °C, fixed with 70% ethanol at 4 °C, suspended in 1 mL of propidium iodide solution (final concentration, 50 μg/mL) containing 100 U/mL RNase A and 0.1% glucose, and incubated for 30 min in the dark [34,35]. Flow cytometry was then performed using a FACSCalibur flow cytometer (FACSDiva7.0, Becton-Dickinson, San Jose, CA, USA) and cell populations were identified using the Cell Quest software (Becton-Dickinson). Red fluorescence, indicative of propidium iodide uptake by damaged cells, was measured at 585/542 nm using logarithmic amplification. Electronic compensation was performed to account for the spectral overlap [34,35].

### 2.5. Immunofluorescence Analysis

PT and OR CRC cells were treated with PBS or oxaliplatin (2.5 μM) for 24 h, fixed with 4% paraformaldehyde (pH 7.4) at room temperature for 10 min. The cells were then permeabilized with 0.1% Triton X-100 (Sigma-Aldrich) for 10 min at room temperature. Cells were washed thrice with DPBS and incubated overnight at 4 °C with LC3B antibody diluted to 1:50. After washing thrice with DPBS, the cells were incubated with Alexa Fluor 488-conjugated secondary antibody diluted to 1:100 (Invitrogen) at room temperature for 1 h and washed thrice with DPBS. The cells were carefully mounted on a slide with a coverslip using a mounting medium containing DAPI. The images were analyzed using a confocal microscope (Nikon, Minato City, Japan).

To investigate the effects of AMPK on autophagy in ST and OR CRC cells, ST and OR CRC cells were treated with PBS, 20 mM metformin or 10 μM compound C for 24 h. The remaining procedures were the same as those described above.

### 2.6. Protein Isolation and Immunoblot Analysis

To evaluate protein expression, PT, ST and OR CRC cells were washed thrice with ice-cold DPBS and harvested with cell lysis buffer supplemented with a protease inhibitor cocktail. Proteins were quantified using a BCA assay kit (Pierce, Rockford, IL, USA), according to the manufacturer’s protocol. Proteins (20–40 μg) were resolved on a 7.5–15% SDS-PAGE and transferred to nitrocellulose membranes (Pall Corp., Ann Arbor, MI, USA). The blots were incubated with appropriate primary antibodies diluted to 1:1000 or GAPDH diluted to 1:10,000 at 4 °C overnight. GAPDH was used as a loading control.

To investigate the effects of AMPK on protein expression involved in autophagy and glycolysis in ST and OR CRC cells, ST and OR CRC cells were treated with 20 mM metformin or 10 μM compound C for 24 h. The remaining procedures were the same as those described above.

### 2.7. Statistical Analysis

Data are presented as mean ± standard deviation (SD). Significance of the differences was analyzed using or Tukey’s post-test for comparison among three means or more after analysis of variance (ANOVA) or Student’s *t*-test between two unpaired data using GraphPad Prism 5.0 software. Statistical significance was set at *p* < 0.05.

## 3. Results

### 3.1. Establishment of Resistance to Oxaliplatin in CRC Cells 

Resistance to oxaliplatin was induced in human CRC cells, HCT116, HT29, SW480 and SW620 via chronic exposure to oxaliplatin with a stepwise increase in its concentration to a maximum of 2.5 μM over approximately 5–6 months. The cell morphology was slightly altered and the growth rates of OR CRC cells were slower than those of their respective PT CRC cells. Cell proliferation was inhibited by oxaliplatin in a concentration-dependent manner in both PT and OR CRC cells (Figure 1). The respective IC_50_ values of oxaliplatin in HCT116/OR, HT29/OR, SW480/OR and SW620/OR were significantly increased by 4.40-, 9.19-, 4.96- and 12.7-fold of those of their respective PT CRC cells (Table 1), confirming the successful establishment of resistance to oxaliplatin in these cells.

To examine whether oxaliplatin resistance induced resistance to other chemotherapeutic drugs, both PT and OR CRC cells of HCT116, HT29, SW480 and SW620 were independently treated with DOX, 5-FU and TSA at appropriate concentrations for 72 h. Mean IC_50_ values are listed in Table 1. The inhibition of cell proliferation increased in a concentration-dependent manner in both PT and OR CRC cells, but OR CRC cells exhibited greater chemoresistance than their respective PT CRC cells for DOX; the respective IC_50_ values of DOX for the HCT116/OR, HT29/OR, SW480/OR and SW620/OR CRC cells were approximately 2.84, 1.79, 5.76 and 3.97-fold higher than those of their respective PT CRC cells. HCT116/OR and SW620/OR cells also showed increased resistance to 5-FU by 4.19 and 16.0-fold of those of their respective PT CRC cells, whereas no such effect was observed for HT29/OR and SW480/OR cells (Table 1). The IC50 values of TSA significantly increased by 1.32- and 1.58-fold in HT29/OR and SW620/OR CRC cells, respectively, but the difference was not large compared to other drugs, so the resistance to TSA did not seem to develop in CRC cells (Table 1). 

### 3.2. Effects of Oxaliplatin Resistance on Drug Efflux Pumps

Drug efflux pumps are associated with drug resistance in various types of cancer. In particular, ATP-binding cassette (ABC) transporters are proteins that mediate the efflux of absorbed drugs from CRC cells and are considered to be one of the possible drivers of oxaliplatin resistance. Therefore, P-gp, MRP2 and BCRP protein levels were determined using immunoblotting (Figure 2). The protein expression of P-gp and MRP2 was significantly increased in HCT116/OR, HT29/OR, SW480/OR and SW620/OR CRC cells compared to their respective PT CRC cells (Figure 2A,B). The expression of BCRP was increased in SW480/OR and SW620/OR CRC cells (Figure 2A,B). Therefore, ABC transporters, especially P-gp and MRP2, appear to mediate cellular resistance to oxaliplatin and other anticancer drugs in OR CRC cells.

### 3.3. Effects of Oxaliplatin Resistance on Cell Cycle Progression 

Oxaliplatin has been reported to induce cell cycle arrest and apoptosis in cancer cells [25,28]. Flow cytometry was performed to assess the effects of oxaliplatin resistance on cell cycle progression in PT and OR CRC cells. As shown in Figure 3A,B, the patterns of cell cycle progression among the four CRC cell lines were similar except for SW620 CRC cells. G_2_/M phase arrest was observed in HT29/ST and SW480/ST CRC cells. However, S phase arrest was observed in the HCT116/ST, SW480/ST and SW620/ST CRC cells (Figure 3A,B). In addition, sub-G_1_ peaks were significantly increased in the HT29/ST and SW620/ST CRC cells, indicating that oxaliplatin was able to drive apoptosis in these cells. The proportion of cell populations at each phase in the OR CRC cells did not respond to oxaliplatin treatment and was maintained at levels similar to those in PT CRC cells except SW620/OR CRC cells (Figure 3A,B).

The immunoblot analysis supported these results. The expression of cleaved PARP was increased in the HCT116/ST, HT29/ST, SW480/ST and SW620/ST CRC cells compared to their respective PT CRC cells (Figure 3C,D), which was consistent with the results of flow cytometry (Figure 3A,B). In addition, the expression of pro-apoptotic proteins, Bad increased in ST CRC cells but decreased in OR CRC cells except SW480/OR. The expression of pro-apoptotic protein, Bax increased in HCT116/ST and HT29/ST CRC cells but decreased in HCT11/OR and SW480/OR CRC cells (Figure 3C,D). The expression of p53 was significantly increased in all ST CRC cells compared to that in their respective PT CRC cells, and it was restored to the basal level or decreased in the OR CRC cells. The expression of cyclin D1 was decreased in ST CRC cells; however, it returned to the basal levels of their respective PT CRC cells in OR CRC cells, except for HCT116/OR (Figure 3C). The expression of CDK4 was significantly increased in HCT116/ST, SW480/ST and SW620/ST CRC cells compared to each of their PT CRC cells (Figure 3C,D) 

### 3.4. Effects of Oxaliplatin Resistance on Autophagy 

To determine the effects of oxaliplatin resistance on the induction of autophagy, the expression of LC3B, an important constituent of the autophagosomal membrane [36], was analyzed using immunofluorescence staining to visualize the expression of LC3B (Figure 4A). The expression of LC3B increased in ST CRC cells compared to that in their respective PT CRC cells but decreased in OR CRC cells. These results suggested that autophagy was induced in ST CRC cells by a single exposure to oxaliplatin but gradually disappeared after chronic exposure to oxaliplatin in OR CRC cells. The induction of autophagy was confirmed by immunoblot analysis of LC3B (Figure 4B,C). When activated, LC3B-I forms conjugate with phosphoethanolamine to produce LC3B-II, which is incorporated to autophagosomal membrane and is involved in autophagosomal degradation [37,38]. The expression of LC3B-II was increased in ST CRC cells but was restored to the basal level of PT CRC cells in their respective OR CRC cells. However, the expression of Beclin-1, a factor involved in the initiation phase of autophagy [36], was comparable among the PT, ST and OR CRC cells except HCT116/OR CRC cells (Figure 4B,C). 

### 3.5. Effect of Oxaliplatin Resistance on AMPK and Akt/mTOR Signaling Pathway

To explore whether the autophagy induced by a single exposure to oxaliplatin was related to AMPK and/or Akt/mTOR signaling pathways, immunoblot analysis was performed in PT, ST and OR CRC cells of HCT116, HT29, SW480 and SW620 (Figure 5). The protein levels of phospho-AMPK (Thr172) and its negative regulator, phospho-Akt (Ser473), were downregulated in ST CRC cells compared to those in their respective PT CRC cells and increased in OR CRC cells. The levels of phospho-mTOR (Ser2448), a downstream target of both AMPK and Akt, were decreased or comparable in ST CRC cells but increased in OR CRC cells compared to those in their respective PT CRC cells. In addition, phospho-p70S6K (Ser424/Thr421), phospho-p70S6K (Thr389) and phospho-S6RP were also overexpressed in the OR CRC cells as compared to their respective PT CRC cells. These observations collectively indicated that oxaliplatin resistance was regulated by the AMPK and/or Akt/mTOR signaling pathway. 

### 3.6. Effects of Oxaliplatin Resistance on the Metabolism of Glucose

To investigate whether the decrease in phospho-AMPK in ST CRC cells and restoration or increase in phospho-AMPK in OR CRC cells were related to glucose metabolism, the expression of the enzymes involved in glycolysis were determined by immunoblot analyses. The expression of the proteins involved in glycolysis, GLUT1, HK2 and PFK1 decreased in ST CRC cells and increased in OR CRC cells relative to their respective ST CRC cells except that GLUT1 in SW480/OR and HK2 in SW620/OR were decreased compared to their respective PT and/or ST CRC cells (Figure 6A,B). The expression patterns of GLUT1, PFKFB3 and PFK1 were similar to those of phospho-AMPK in the PT, ST and OR CRC cells. The protein expression of PFKFB3, a downstream target of AMPK and regulator of PFK1, was also activated by AMPK in OR CRC cells, followed by the activation of PFK1. These results indicated that AMPK phosphorylation activated the enzymes involved in glycolysis to induce the energy levels available for utilization by OR CRC cells to sustain their proliferation and growth. The expression of PKM2 was significantly decreased in HT29/ST but increased in SW480/OR CRC cells compared to their each PT CRC cells (Figure 6A,B). The protein expression of LDHA remained comparable among the PT, ST and OR CRC cells.

### 3.7. Effects of AMPK on Autophagy and Glucose metabolism

To confirm whether AMPK is involved in the regulation of autophagy and glucose metabolism, ST and OR CRC cells were treated with 10 or 20 mM metformin, an AMPK activator, and 10 μM compound C, an AMPK inhibitor, respectively. As shown in Figure 7A, co-treatment with oxaliplatin and metformin did not show any synergistic effect on the inhibition of PT CRC cell proliferation. In contrast, a co-treatment with oxaliplatin and compound C showed significantly increased the inhibition of PT CRC cell proliferation (Figure 7A). In OR CRC cells, co-treatment with metformin or compound C showed synergistic effect on the inhibition of OR CRC cell proliferation and co-treatment with compound C showed significantly greater inhibition of cell proliferation compared to co-treatment with metformin in OR CRC cells (Figure 7B). In addition, the induced autophagy in ST CRC cells (Figure 4A) was decreased by a co-treatment with metformin but was significantly increased after co-treatment with compound C (Figure 8A). Similar results were also obtained from OR CRC cells (Figure 8B). 

Analysis of the protein expression also confirmed the above results that phospho-AMPK was increased and LC3B-II was decreased by metformin treatment in ST CRC cells, along with an increase in phospho-Akt (Figure 9A). Metformin also increased the protein expression of GLUT1, PFKFB3 and PFK1 in ST CRC cells (Figure 9A). On the contrary, phospho-AMPK was decreased and LC3B-II was increased by compound C treatment in ST CRC cells, along with a slight decrease in phospho-Akt (Figure 9B). Compound C also decreased the protein expression of GLUT1 and PFK1 in ST CRC cells and PFKFB3 in SW480/ST and SW620/ST cells (Figure 9B). Co-treatment with metformin increased phospho-AMPK in HCT116/OR and HT29/OR and phospho-Akt in HCT116/OR and SW480/OR cells and the other proteins were comparable or slightly decreased in OR CRC cells (Figure 9C). On the contrary, phospho-AMPK was decreased by compound C treatment, and the expression of LC3B-II was increased by a decrease in phospho-Akt (Figure 9D). Furthermore, the protein expression levels of GLUT1, PFKFB3 and PFK1 were decreased by treatment with compound C in OR CRC cells (Figure 9D). Therefore, inactivation of AMPK appeared to improve oxaliplatin sensitivity in both ST and OR CRC cells through induction of autophagy.

## 4. Discussion

Oxaliplatin presents anticancer activity by intercalating DNA and blocking both replication and transcription in CRC [39]. However, repeated exposure to oxaliplatin can lead to chemotherapy failure owing to anticancer drug resistance. In this study, we generated OR CRC cells by gradually increasing the concentration of oxaliplatin. The significant effects on cell proliferation and G_2_/M phase arrest observed in ST CRC cells gradually disappeared on chronic exposure to oxaliplatin and resulted in increased IC_50_ values by 4.40~12.7 fold and restoration of G_2_/M phase to the level of their respective untreated PT CRC cells in the OR CRC cells, confirming the development of resistance to oxaliplatin. 

Based on oxaliplatin resistance in the OR CRC cells, we suggest the following mechanisms for oxaliplatin resistance in OR CRC. First, the protein expression of P-gp was increased in the OR CRC cells of HCT116, HT29, SW480 and SW620 compared to their respective PT CRC cells. This suggests that as a substrate of P-gp, oxaliplatin that enters the cells is easily excreted by P-gp. This indicates that overexpression of P-gp results in a resistance to oxaliplatin, validating the increase in the IC_50_ values of oxaliplatin for OR CRC cells. This was also evident through a significant increase in the IC_50_ values of DOX for OR CRC cells of HCT116, HT29, SW480 and SW620. However, the IC_50_ values of 5-FU and TSA, which are not substrates of P-gp, were comparable between PT and OR CRC cells, except those of 5-FU in HT29/OR and SW480/OR CRC cells, indicating that P-gp may be partially involved in oxaliplatin resistance in OR CRC cells. The induction of chemoresistance by P-gp upregulation has been observed not only with oxaliplatin but also with DOX and paclitaxel [40]. Butyrate resistant (BR) HCT116 CRC cells have also been reported to present resistance to DOX and paclitaxel [41] in response to P-gp overexpression at higher butyrate concentrations (data not shown).

Second, as shown in Figure 3C, oxaliplatin resistance in OR CRC cells might be caused by decreased expression of apoptotic and/or tumor-suppressing proteins, such as PARP, Bad, Bax and p53, or increased expression of tumorigenic proteins, such as cyclin D1. This is in contrast to the observations in ST CRC cells that showed decreased expression of tumorigenic proteins and increased expression of pro-apoptotic proteins in response to oxaliplatin. Therefore, oxaliplatin resistance might be attributed to changes in the expression of pro-apoptotic, tumor-suppressing and/or tumorigenic proteins, as well as the increased expression of P-gp in OR CRC cells. Similar results have also been reported in BR human CRC cells, where Bim and Bax levels were decreased in HCT116/BR CRC cells [41] and the apoptotic pathway was impaired through the inactivation of Bax and upregulation of Bcl-2 in human BCS-TC2/BR CRC cells [42]. Upregulation of cyclin D1 and downregulation of p21 and Bax are also involved in BR human CRC cells, HCT116, HT29 and SW480 [35]. HeLa human cervical cancer cells have also been reported to develop resistance to butyrate via upregulation of cyclin D1 [43]. Tamoxifen-resistant breast cancer cells, MCF7 and T47D, have been reported to overexpress cyclin D1 owing to the inhibition of cyclin D1 degradation [44]. 

Along with these results, autophagy was induced via upregulation of LC3B-II in ST CRC cells. The protein expression of LC3B-II was higher in ST CRC cells than that in their respective PT CRC cells and decreased in OR CRC cells, suggesting that the autophagy observed in ST and OR CRC cells has a tumor suppressive role. Oxaliplatin treatment in PT CRC cells inhibited their proliferation by decreasing phospho-Akt, followed by decreasing phospho-mTOR and other downstream molecules, such as phospho-p70S6K and phospho-S6RP, resulting in the induction of tumor-suppressive autophagy in ST CRC cells. However, the Akt/mTOR signaling pathway was significantly activated to protect OR CRC cells and resulted in a reduction of autophagy in HCT116/OR, HT29/OR, SW480/OR and SW620/OR CRC cells. Reduced autophagy has also been observed in 5-FU resistant human SNUC5 CRC cells [45]. However, BR CRC cells showed an increase in autophagy mediated by the activation of Akt/mTOR signaling pathway compared to their respective PT CRC cells, suggesting that autophagy in BR CRC cells appears to be tumor-protective and therefore, supports BR CRC cell survival [35]. 

Akt is a negative regulator of AMPK [18,46]. Akt inhibits the phosphorylation of AMPK at Thr172 and blocks AMPK activation [46,47]. Although the expression of phospho-Akt was decreased in ST CRC cells, the level of phospho-AMPK was also significantly reduced but autophagy was induced. In contrast, the expression of both phospho-Akt and phospho-AMPK was restored or increased, and autophagy was inhibited in OR CRC cells, suggesting that the autophagy observed in ST and OR CRC cells was independent of AMPK. Several studies have reported that the association between AMPK and Akt is bidirectional, and AMPK can induce the phosphorylation of Akt and enhance its activity [18,46]. Co-treatment with metformin activated AMPK in ST CRC cells; however, this observation was not accompanied by the induction of autophagy due to the activation of the Akt/mTOR pathway. Similarly, co-treatment with compound C inhibited AMPK and induced autophagy through the induction of LC3B-II in ST CRC cells. Co-treatment with metformin did not increase autophagy in OR CRC cells but co-treatment with compound C inhibited AMPK and induced autophagy due to the inhibition of Akt in OR CRC cells. Compound C has been reported to inhibit both AMPK and Akt and induce autophagy in L929 fibrosarcoma, B16 melanoma and C3 glioma cells [48]. Therefore, autophagy is also induced by treatment with the PI3K/Akt inhibitor, LY294002 or mTOR inhibitor, rapamycin [48]. Overall, oxaliplatin-induced autophagy in CRC cells is AMPK-independent and principally regulated by the Akt/mTOR pathway. Similar results were reported in a previous study showing that the inhibition of AMPK, Akt and Erk pathways induced cell death in DOX-resistant ovarian cancer cells [49].

AMPK is activated under conditions of metabolic stress such as starvation and is known to modulate the Warburg effect [50]. Activation of AMPK in OR CRC cells appears to be related to glucose utilization by OR CRC cells to maintain survival and is independent of autophagy. In OR CRC cells, enzymes involved in glycolysis were restored or increased alongside AMPK phosphorylation; especially GLUT1, PFKFB3 and PFK1 were overexpressed compared to their respective PT and/or ST CRC cells. GLUT1 levels are upregulated in cancer cells to facilitate glucose uptake through the phosphorylation of AMPK [51,52]. Therefore, the induction of autophagy in OR CRC cells treated with compound C seemed to be associated with decreased phospho-AMPK and GLUT1 levels. Similar results were also reported in p53-mutant CRC cells [53]. In addition, inhibition of GLUT1 by silybin has been shown to counteract DOX resistance in DOX-resistant LoVo CRC cells [54]. Similarly, GLUT1 inhibition by phloretin has been associated with sensitization of cancer cells to daunorubicin and overcome drug resistance in SW620 CRC cells under hypoxia [55]. Furthermore, AMPK phosphorylation appears to regulate PFK1 indirectly. AMPK appeared to activate PFKFB3, followed by an increase in the level of fructose-2,6-bisphosphate, which in turn activates PFK1, the rate-limiting glycolytic enzyme [56]. Another study has reported that AMPK activation increases glucose uptake, followed by an increase in lactate production by AMPK-dependent phosphorylation of PFKFB3 during mitotic arrest in breast cancer cells [57]. Inhibition or knockdown of AMPK or induction of autophagy in breast cancer cells has been reported to enhance cell death during the mitotic phase, therefore improving the antitumor efficiency [57]. In addition, knockdown of AMPK in a xenograft model prepared from MDA-MB-231 human breast cancer cells showed a reduction in lactate production, thus, inhibiting the cancer cell growth [51]. Taken together, AMPK may play a pivotal role in glucose metabolism in cancer cells. Therefore, targeting AMPK may inhibit the Warburg effect in cancer cells, allowing therapeutic resistance to be overcome by increasing chemosensitivity.

## 5. Conclusions

An illustrative summary of this study is presented in Figure 10. In PT CRC cells, a single treatment with oxaliplatin caused a decrease in both the Akt/mTOR signaling pathway and AMPK phosphorylation, inducing tumor-suppressive autophagy and anticancer effects. In contrast, the phospho-AMPK and Akt/mTOR signaling pathways were activated in OR CRC cells, reducing tumor-suppressive autophagy, which protects these cells by increasing the levels of enzymes involved in glycolysis, especially GLUT1, PFKFB3 and PFK1, thereby maintaining resistance to oxaliplatin. Inhibition of AMPK in OR CRC cells induces tumor-suppressive autophagy through inactivation of Akt/mTOR pathway and decrease in the level of GLUT1, PFKFB3 and PFK1. Collectively, autophagy observed in ST and OR CRC cells is AMPK-independent and principally regulated by the Akt/mTOR pathway. Activation of AMPK plays a role in oxaliplatin resistance by enhancing glycolysis for OR CRC cell survival. Therefore, targeting AMPK may provide solutions to overcome chemoresistance in OR CRC cells and restore chemosensitivity to anticancer drugs, including oxaliplatin.

## Figures and Tables

**Figure 1 biomedicines-10-02690-f001:**
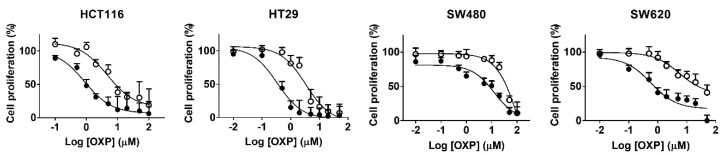
Effects of oxaliplatin on proliferation of the colorectal cancer (CRC) cells. Cell proliferation was estimated by MTT assay in parental (PT, ●) and oxaliplatin-resistant (OR, ○) CRC cells of HCT116, HT29, SW480 and SW620. Cells were treated with various concentration of oxaliplatin for 72 h. The data are expressed as mean ± standard deviation of triplicate experiments. The *x*-axis represents log scale.

**Figure 2 biomedicines-10-02690-f002:**
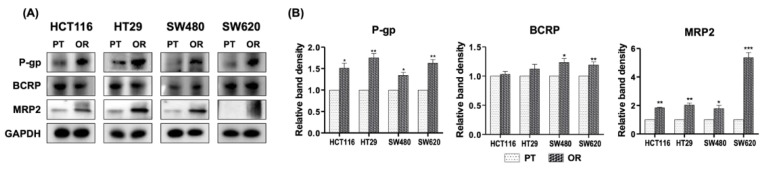
(**A**) Effects of oxaliplatin resistance on the expression of drug efflux pumps. Expression of P-gp and BCRP in parental (PT) and oxaliplatin-resistant (OR) colorectal cancer cells of HCT116, HT29, SW480 and SW620 was determined by immunoblot analysis. (**B**) Graphical analysis of band intensities normalized for protein loading with GAPDH. Columns and error bars represent means ± standards deviation of immunoblot band intensities of 3 separate protein preparations. * *p* < 0.05; ** *p* < 0.01; *** *p* < 0.001. P-gp, P-glycoprotein; BCRP, breast cancer resistance protein; MRP2, multidrug resistance-associated protein 2; GAPDH, glyceraldehyde 3-phosphate dehydrogenase.

**Figure 3 biomedicines-10-02690-f003:**
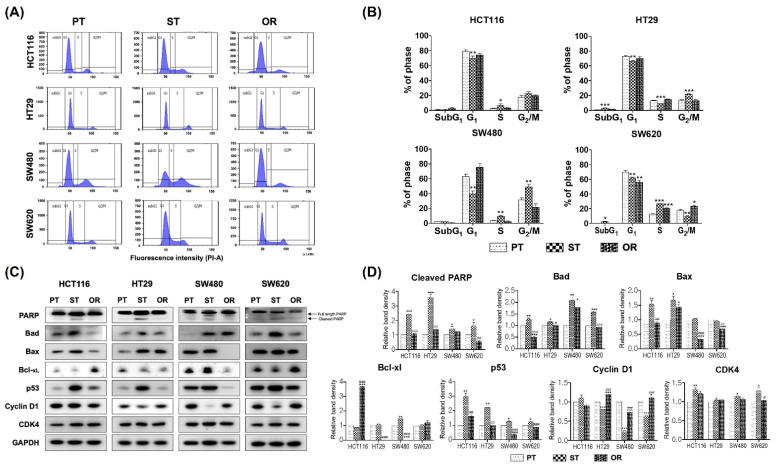
Effects of oxaliplatin resistance on the cell cycle progression and induction of apoptosis in the parental (PT), 2.5 μM oxaliplatin single treated PT for 24 h (ST) and oxaliplatin-resistant (OR) colorectal cancer (CRC) cells of HCT116, HT29, SW480 and SW620. (**A**) Cellular DNA was analyzed by flow cytometry. (**B**) The percentage of cells in each phase was calculated from each histogram. (**C**) Protein expression involved in apoptosis and cell cycle progression was analyzed by immunoblot analysis. (**D**) Graphical analysis of band intensities normalized for protein loading with GAPDH. Columns and error bars represent means ± standards deviation of immunoblot band intensities of 3 separate protein preparations. The *** statistics showed the comparison with PT and the ^#^ statistics showed the comparison between ST and OR. * or ^#^, *p* < 0.05; ** or ^##^, *p* < 0.01; *** or ^###^, *p* < 0.001. GAPDH, glyceraldehyde 3-phosphate dehydrogenase.

**Figure 4 biomedicines-10-02690-f004:**
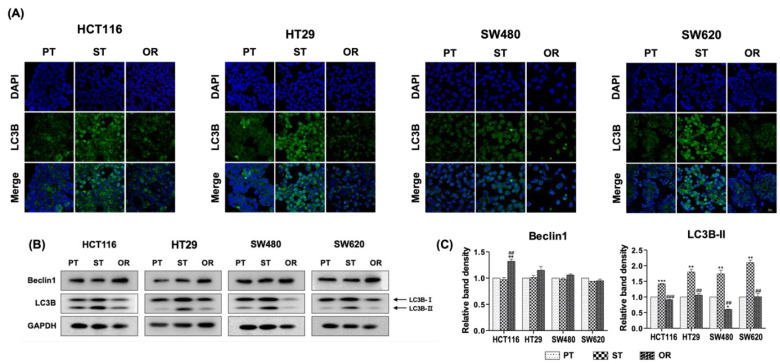
Effects of oxaliplatin resistance on the induction of autophagy in the parental (PT), 2.5 μM oxaliplatin single treated PT for 24 h (ST) and oxaliplatin-resistant (OR) colorectal cancer (CRC) cells of HCT116, HT29, SW480 and SW620. (**A**) LC3B was visualized by immunofluorescence using confocal microscopy. LC3B protein was labeled green (Alexa 488) and the nucleus was stained with DAPI. The size of scale bar was 20 μm. (**B**) Protein expression of Beclin-1 and LC3B was determined by immunoblot analysis. (**C**) Graphical analysis of band intensities normalized for protein loading with GAPDH. Columns and error bars represent means ± standards deviation of immunoblot band intensities of 3 separate protein preparations. The *** statistics showed the comparison with PT and the ^#^ statistics showed the comparison between ST and OR. * *p* < 0.05; ** or ^##^, *p* < 0.01; *** or ^###^, *p* < 0.001. GAPDH, glyceraldehyde 3-phosphate dehydrogenase.

**Figure 5 biomedicines-10-02690-f005:**
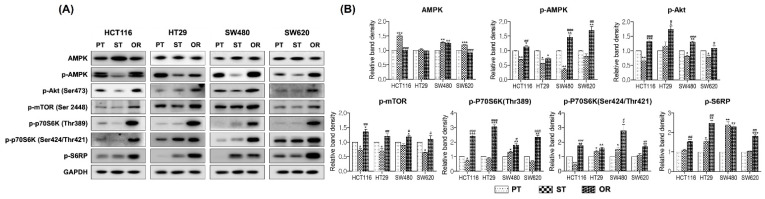
Effect of oxaliplatin resistance on AMPK and Akt/mTOR signaling pathway in the parental (PT), 2.5 μM oxaliplatin single treated PT for 24 h (ST) and oxaliplatin-resistant (OR) colorectal cancer (CRC) cells of HCT116, HT29, SW480 and SW620. (**A**) Protein expression was determined by immunoblot analysis. (**B**) Graphical analysis of band intensities normalized for protein loading with GAPDH. Columns and error bars represent means ± standards deviation of immunoblot band intensities of 3 separate protein preparations. The * statistics showed the comparison with PT and the ^#^ statistics showed the comparison between ST and OR. * or ^#^, *p* < 0.05; ** or ^##^, *p* < 0.01; *** or ^###^, *p* < 0.001. p-, phospho-; GAPDH, glyceraldehyde 3-phosphate dehydrogenase.

**Figure 6 biomedicines-10-02690-f006:**
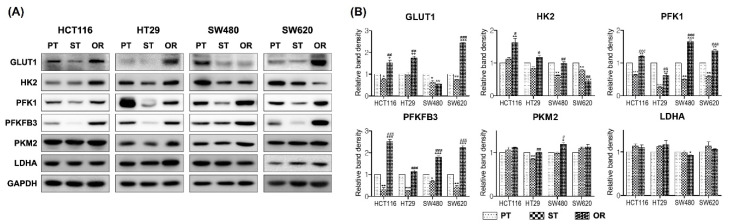
Effect of oxaliplatin resistance on enzymes involved in the glycolysis in the parental (PT), 2.5 μM oxaliplatin single treated PT for 24 h (ST) and oxaliplatin-resistant (OR) colorectal cancer (CRC) cells of HCT116, HT29, SW480 and SW620. (**A**) Protein expression was determined by immunoblot analysis. (**B**) Graphical analysis of band intensities normalized for protein loading with GAPDH. Columns and error bars represent means ± standards deviation of immunoblot band intensities of 3 separate protein preparations. The * statistics showed the comparison with PT and the ^#^ statistics showed the comparison between ST and OR. * or ^#^, *p* < 0.05; ** or ^##^, *p* < 0.01; *** or ^###^, *p* < 0.001. GLUT1, glucose transporter 1; HK2, hexokinase 2; PFK1, phosphofructokinase 1; PKM2, pyruvate kinase M2; LDHA, lactate dehydrogenase A; PFKFB3, 6-phosphofructo-2-kinase/fructose-2,6-biphosphatase 3; GAPDH, glyceraldehyde 3-phosphate dehydrogenase.

**Figure 7 biomedicines-10-02690-f007:**
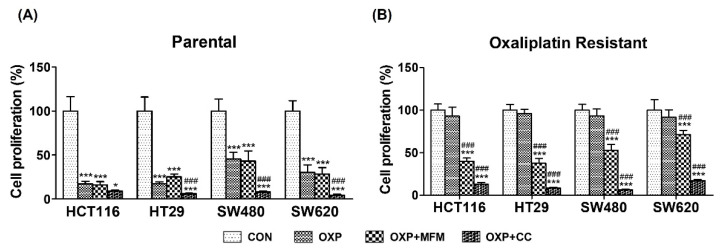
Effects of AMPK on the proliferation of HCT116, HT29, SW480 and SW620 colorectal cancer (CRC) cells by treating metformin as an AMPK activator or compound C as an AMPK inhibitor. (**A**) Parental (PT) CRC cells were pretreated with or without 10 mM metformin and 10 μM compound C followed by 2.5 μM oxaliplatin for 72 h. (**B**) Oxaliplatin resistant (OR) CRC cells were pretreated with or without 10 mM metformin and 10 μM compound C followed by 2.5 μM oxaliplatin for 72 h. The cell proliferation was determined by an MTT assay. The data are expressed as mean ± standard deviation of triplicate experiments. * *p* < 0.05; *** or ^###^, *p* < 0.001. The * statistics showed only the comparison with CON and the ^#^ statistics showed the comparison with OXP. CON, control; AMPK, AMP-activated protein kinase; MFM, metformin; CC, compound C; OXP, oxaliplatin.

**Figure 8 biomedicines-10-02690-f008:**
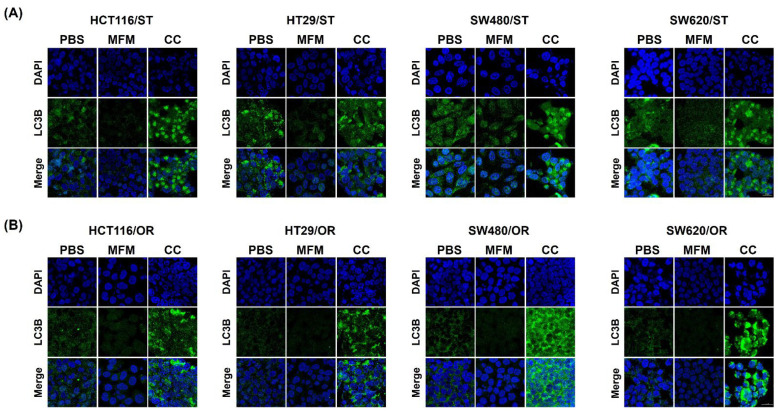
Effects of AMPK on the induction of autophagy in HCT116, HT29, SW480 and SW620 colorectal cancer (CRC) cells by treating metformin as an AMPK activator or compound C as an AMPK inhibitor. (**A**) Parental (PT) CRC cells were pretreated with or without 20 mM metformin and 10 μM compound C followed by 2.5 μM oxaliplatin for 24 h. (**B**) Oxaliplatin-resistant (OR) CRC cells were pretreated with or without 20 mM metformin and 10 μM compound C followed by 2.5 μM oxaliplatin for 24 h. LC3B was visualized by immunofluorescence using confocal microscopy. LC3B protein was labeled green (Alexa 488) and the nucleus was stained with DAPI. The size of scale bar was 20 μm. This experiment was performed in three independent replicates. PBS, phosphate-buffered saline; AMPK, AMP-activated protein kinase; MFM, metformin; CC, compound C.

**Figure 9 biomedicines-10-02690-f009:**
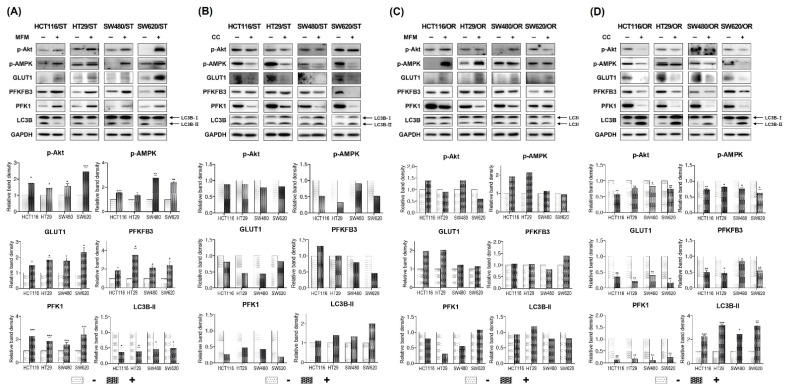
Effects of AMPK on the protein expression involved in glycolysis and autophagy in HCT116, HT29, SW480 and SW620 colorectal cancer (CRC) cells by treating metformin as an AMPK activator or compound C as an AMPK inhibitor. (**A**) Parental (PT) CRC cells were pretreated with or without 20 mM metformin followed by 2.5 μM oxaliplatin for 24 h. (**B**) Parental (PT) CRC cells were pretreated with or without 10 μM compound C followed by 2.5 μM oxaliplatin for 24 h. (**C**) Oxaliplatin resistant (OR) CRC cells were pretreated with or without 20 mM metformin followed by 2.5 μM oxaliplatin for 24 h. (**D**) Oxaliplatin resistant (OR) CRC cells were pretreated with or without 10 μM compound C followed by 2.5 μM oxaliplatin for 24 h. Protein expression was determined by immunoblot analysis. Graphical analysis of band intensities normalized for protein loading with GAPDH. Columns and error bars represent means ± standards deviation of immunoblot band intensities of 3 separate protein preparations. * *p* < 0.05; ** *p* < 0.01; *** *p* < 0.001. p-, phospho-; GLUT1, glucose transporter 1; AMPK, AMP-activated protein kinase; PFK1, phosphofructokinase 1; PFKFB3, 6-phosphofructo-2-kinase/fructose-2,6-biphosphatase 3; GAPDH, glyceraldehyde 3-phosphate dehydrogenase; MFM, metformin; CC, compound C.

**Figure 10 biomedicines-10-02690-f010:**
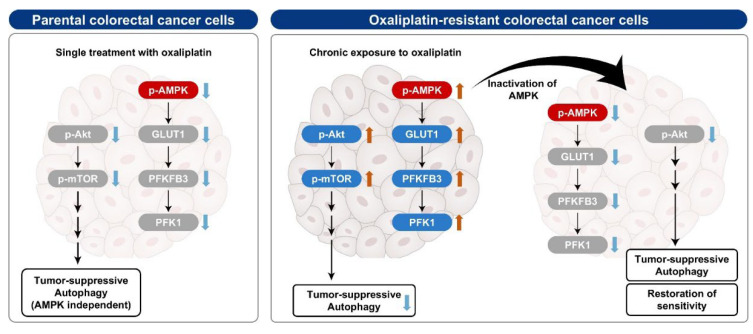
Summary of possible mechanism in parental (PT) and oxaliplatin-resistant (OR) colorectal cancer cells. p-, phospho-.

**Table 1 biomedicines-10-02690-t001:** Mean IC_50_ ± standard deviation values (*n* = 3) of oxaliplatin, DOX, 5-FU and TSA in PT and OR human colorectal cancer cells of HCT116, HT29, SW480 and SW620.

Drugs	HCT116		HT29		SW480		SW620	
	PT	OR	PT	OR	PT	OR	PT	OR
Oxaliplatin (μM)	1.04 ± 0.065	4.58 ± 0.922 ***	0.446 ± 0.036	4.14 ± 0.986 ***	7.11 ± 0.984	35.3 ± 4.42 ***	0.485 ± 0.319	6.15 ± 4.38 *
DOX (nM)	152 ± 11.8	471 ± 24.0 ***	71.6 ± 5.85	128 ± 10.4 ***	21.7 ± 6.58	125 ± 24.6 ***	31.5 ± 9.79	125 ± 18.6 ***
5-FU (μM)	3.70 ± 0.523	15.4 ± 0.170 **	6.44 ± 0.470	9.03 ± 2.81	7.56 ± 1.25	7.94 ± 0.551	1.26 ± 0.447	20.2 ± 5.96 ***
TSA (nM)	97.3 ± 9.18	102 ± 6.09	93.1 ± 7.65	123 ± 10.8 ***	93.9 ± 10.7	105 ± 9.33	105 ± 8.60	166 ± 2.60 ***

IC_50_, inhibitory concentration of cell growth by 50%; DOX, doxorubicin; 5-FU, 5-fluorouracil; TSA, trichostatin A; PT, parental; OR, oxaliplatin-resistant. * *p* < 0.05; ** *p* < 0.01; *** *p* < 0.001.

## Data Availability

All data described in the study can be found in the article and we do not have any supporting data.

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
