# Peer review of "Role of AMPK in Regulation of Oxaliplatin-Resistant Human Colorectal Cancer"

_biomedicines, 2022, doi:10.3390/biomedicines10112690_

Round 1

Reviewer 1 Report

In this study, Park et al. investigated the mechanism of oxaliplatin resistance in cultured colorectal cancer cells.  The authors examined the effects of oxaliplatin resistance on AMPK and Akt-mTOR signaling associated with autophagy and glycolytic metabolism in oxaliplatin-resistant (OR) or the parental (PT) cells under the conditions with oxaliplatin.  They found that oxaliplatin suppressed Akt-mTOR signaling and AMPK phosphorylation, resulting in the induction of tumor-suppressing autophagy in PT cells.  In contrast, Akt-mTOR signaling and AMPK phosphorylation were activated by oxaliplatin in OR cells.  Regarding tumor-suppressing autophagy, the levels of enzymes involved in glycolysis (e.g., GLUT1, PFKFB3, and PFK1) were up-regulated in OR cells but not PT cells.  Taken together, the authors concluded that AMPK signaling plays a critical role in oxaliplatin resistance by enhancing the glycolysis-related pathway for cell survival in OR colorectal cancer cells.

This study contains novel and interesting findings of the AMPK-related mechanism of oxaliplatin resistance in cultured colorectal cancer cells.  Although it demonstrates several meaningful findings of AMPK signaling altered by oxaliplatin resistance, there are several concerns below to be addressed.     

Major concerns.

Figs. 2, 3C, 4B, 5, 6, and 9: For all immunoblots, the band intensities should be quantified by densitometry, and quantitative data should be presented to show changes in band intensities.

Minor concerns.

Page 10, line 351: “PKF1” should be PFK1.

Page 10, line 354: “PKF1” should be PFK1.

Author Response

Response to Reviewer 1

The authors investigated DDI of tofacitinib with voriconazole in vitro assays and in vivo. Overall, the manuscript is clearly addressed the DDI mechanism between two drugs. However, a few flaws are found across the manuscript and should be revised for publication.

 Major comments

  1. Figs. 2, 3C, 4B, 5, 6 and 9: for all immunoblots, the band intensities should be quantified by densitometry and quantitative data whould be presented to show changes in band intensities.

Response

As the reviewer pointed out, immunoblot band intensities were quantified using ImageJ and was represented as bar graphs with statistics in Figs. 2, 3C, 4B, 5, 6 and 9.

Minor comments

  1. Page 10, line 351: “PKF1” should be “PFK1”. Page 10, line 354: “PKF1” should be “PFK1”.

Response

“PKF1” was corrected to “PFK1” in page 10, line 371 and 374.

Reviewer 2 Report

Dear authors,

The article has been well organized and the design of the study is well defined.

However, some results should be described more broadly according to what is observed in the figures. Sometimes, the figures of western analysis do not reflect what is described in the results. Bar graphs representing protein levels and statitics, in addition to the western image, should be added to the figures.

 In addition, to draw conclusions from the figures 7, 8 and 9 it is necessary to obtain results after treatment with metformin and CC in both lines (PT and OR in figure 7; and ST and OR in figures 8 and 9). These data are very important for drawing accurate conclusions about the role of AMPK on autophagy and glucose metabolism in resistance to oxaliplatin.  

I reiterate that the manuscript is well-written and organized, but additional experiments are needed.

All requests and comments have been included as notes in the attached pdf.

Author Response

Response to Reviewer 2

 Major comments

  1. The article has been well organized and the design of the study is well defined.However, some results should be described more broadly according to what is observed in the figures. Sometimes, the figures of western analysis do not reflect what is described in the results. Bar graphs representing protein levels and statitics, in addition to the western image, should be added to the figures.

Response

As the reviewer pointed out, immunoblot band intensities were quantified using ImageJ and mean ± standard deviation was represented with statistics and added in Figs. 2, 3C, 4B, 5, 6 and 9 as bar graphs.

  1. In addition, to draw conclusions from the figures 7, 8 and 9 it is necessary to obtain results after treatment with metformin and CC in both lines (PT and OR in figure 7; and ST and OR in figures 8 and 9). These data are very important for drawing accurate conclusions about the role of AMPK on autophagy and glucose metabolism in resistance to oxaliplatin.  

Response

As the reviewer pointed out, ST and OR CRC cells were additionally treated with compound C and metformin, respectively, and additional cell proliferation assay, examination of autophagy and immunoblot analysis were performed. The result for each experiment was explained in the “Results” section as well as added in Figures 7, 8 and 9. This appears in minor comments.

Minor comments

Minor comments are explained in the order you pointed out in the manuscript of PDF files.

  1. Page 1, line 40-41,

“Drug resistance develops not only by repeated administration of anticancer chemotherapeutics but also by the simultaneous use of various anticancer drugs.”

Response

The above sentence was deleted and revised to "The current paradigm states that combination therapy should be the best treatment option because it should prevent the development of drug resistance and be more effective than any one drug on its own" as a reviewer recommends. (Page 1, line 40-42)

  1. Page 2, Add references

Response

References were added throughout the manuscript.

  1. Page 4, line 171

Response

The x-axis in Figure 1 is log scale not standard scale. Generally, log scale is used to evaulate dose-response relationship and identify the sigmoid shape. Therefore, -2, -1, 0, 1 and 2 mean 0.01, 0.1, 1, 10 and 100 mM, respectively. “The x-axis represents log scale” is added at the end of Figure 1 legend. (Page 4, line 171-172)

  1. Page 4, line 184, Table 1

Response

Standard deviation with statistics was added in Table 1. (Page 5, line 185-189)

  1. Page 5, line 205, Figure 2

Response

We found the antibody against MRP1 was not good condition and no more left in our lab. Later, we observed MRP2 is overexpressed in OR CRC cells. Therefore, “MRP1 was undetectable in both the cell types” was deleted. Instead, MRP2 band and graphic data of band intensities with statistics were added in Figure 2. The results were revised as follow: “The expression of BCRP was increased in OR CRC cells except HCT116/OR CRC cells (Figure 2A and 2B). Therefore, ABC transporters, especially P-gp and MRP2, appear to mediate ~” and Figure 2 legend was also modified. (Page 5, line 204-2011)**)

  1. Page 6, line 226: Figure 3B “HCT116/ST” in G2/M phase

Response

“HCT116/ST” was deleted. The sentence “G2/M phase arrest was observed in HCT116/ST, HT29/ST and SW480/ST CRC cells” was revised to “G2/M phase arrest was observed in HT29/ST and SW480/ST CRC cells” as the reviewer’s recommendation. (Page 5, line 228)

  1. Page 6, line 227: Figure 3B, SW480/ST in S phase

Response

“HCT116/ST, SW480/ST” was added. The sentence “S phase arrest was observed in the SW620/ST CRC cells” was revised to “S phase arrest was observed in the HCT116/ST, SW480/ST and SW620/ST CRC cells” as the reviewer’s recommendation. (page 6, line 229)

  1. Page 6, line 228: Figure 3B “HCT116/ST, SW480/ST” in subG1 phase

Response

Sub-G1 peaks were significantly increased in the HT29/ST and SW620/ST CRC cells compared to their respective PT CRC cells but were comparable between PT and ST CRC cells of HCT116 and SW480. Therefore, “Sub-G1 peaks were significantly increased in the HT29/ST and SW620/ST CRC cells” is correct. (page 6, line 330-331)

  1. Page 6, line 230: Figure 3B “SW620/OR”

Response

Only SW620/OR CRC cells showed significant changes of cell populations in each cell cycle phase. Therefore, “The proportion of cell populations at each phase in the OR CRC cells did not respond to oxaliplatin treatment and was maintained at levels similar to those in PT CRC cells (Figures 3A and 3B)” was revised to “The proportion of cell populations at each phase in the OR CRC cells did not respond to oxaliplatin treatment and was maintained at levels similar to those in PT CRC cells except SW620/OR CRC cells (Figures 3A and 3B)”. (Page 6, line 332-334)

  1. Page 6, line 235: Graphic bar of immunoblot intensities in Figure 3

Response

The immunoblot band intensities were quantified and bar graph was added with statistics (Figure 3D). (Page 6, line 241-245)

  1. Page 6, line 242: Statistics symbol in Figure 3

Response

The statistics symbols “asterisk” in Figures as well as Figure 3C was enlarged through manuscript. (Page 6)

  1. Page 6, line 245: Cleaved PARP in SW620/ST

Response

We confirme that the band intensity in cleaved PARP of SW620/ST CRC cells was increased compared to that of SW620/PT CRC cells (Figures 3C and 3D). (Page 6, line 247-248)

  1. Page 6, line 248: Bad and Bax protein expression

Response

The sentence “Bad and Bax, increased in ST CRC cells but did not change in OR CRC cells compared to their respective PT CRC cells (Figure 3C).” was revised to “Bad increased in ST CRC cells but decreased in OR CRC cells except SW480/OR. The expression of pro-apoptotic protein, Bax increased in HCT116/ST and HT29/ST CRC cells but decreased in HCT116/OR and SW480/OR CRC cells (Figures 3C and 3D)” as the reviewer’s recommendation. (Page 7, line 250-253)

  1. Page 7, line 263: LC3B-I and LC3B-II

Response

“When activated, LC3B-I forms conjugate with phosphoethanolamine to produce LC3B-II, which is incorporated to autophagosomal membrane and is involved in autophagosomal degradation” was add. (Page 7, line 267-270)

  1. Page 7, line 270: Graphic bar of immunoblot intensities in Figure 4

Response

The immunoblot band intensities were quantified and bar graph was added with statistics (Figure 4C). (Page 7, line 280-284)

  1. Page 8, line 291: Graphic bar of immunoblot intensities in Figure 5.

Response

The immunoblot band intensities were quantified and bar graph was added with statistics (Figure 5B). (Page 8, line 302-305)

  1. Page 8, line 297: Graphic bar of immunoblot intensities in Figure 6

Response

The immunoblot band intensities were quantified and bar graph was added with statistics (Figure 6B). (Page 8, line 311-315)

  1. Page 8, line 308: GLUT1 in SW480OR and HK2 in SW620/OR

Response

“GLUT1, HK2 and PFK1 decreased in ST CRC cells and increased in OR CRC cells relative to their respective PT CRC cells (Figure 6)” was revised to “GLUT1, HK2 and PFK1 decreased in ST CRC cells and increased in OR CRC cells relative to their respective ST CRC cells except that GLUT1 in SW480/OR and HK2 in SW620/OR were decreased compared to their respective PT and/or ST CRC cells (Figures 6A and 6B)” as the reviewer’s recommendation. (Page9, line 322-325).

  1. Page 9, line 319: Figure 7

Response

As the reviewer’s recommendation, ST and OR CRC cells were treated 10 mM metformin and 10 mM compound C, respectively, and the results of cell proliferation were added in Figures 7A and 7B. Legend of Figure 7 was also modified. (Page 9, line 334-342)

  1. Page 9, line 325: CON full names

Response

“CON, control” was added in page 9, line 341-342.

  1. Page 9, line 319: Figure 8

Response

As the reviewer’s recommendation, ST and OR CRC cells were treated 20 mM metformin and 10 mM compound C, respectively, and the results of autophagy were added in Figures 8A and 8B. Legend of Figure 8 was also modified. (Page 10, line 345-353)

  1. Page 10, line 359: Figure 9

Response

As the reviewer’s recommendation, ST and OR CRC cells were treated 20 mM metformin and 10 mM compound C, respectively, and the results of protein expression were added in Figures 9A and 9B. Legend of Figure 9 was also modified. (Page 11, line 388-395)

Round 2

Reviewer 1 Report

The revised paper by Park et al. has been well improved. However, some results are not precisely described based on the data. So, the description of the paper should be revised according to the data results. Several points to be clarified are below.

1. Page 5, lines 201-202: “Resistance to TSA was not observed in any CRC cell line (Table 1)”.

IC50 values of TSA were significantly up-regulated in H129/OR and SW620/OR compared with those of each parental cell (Table 1).

2. Page 5, lines 218-219: “The expression of BCRP was increased in OR CRC cells except HCT116/OR CRC cells (Figure 2A and 2B)”.

BCRP expression was not significantly increased also in HT29/OR (Figure 2B).

3. Page 7, line 256: “No changes in CDK4 expression were observed in any of the four CRC cell lines”.

CDK4 expression was significantly increased in HCR116/ST and SW620/ST compared to each parental cell (Figure 3D).

4. Page 7, lines 269-271: “However, the expression of Beclin-1, a factor involved in the initiation phase of autophagy [36], was comparable among the PT, ST and OR CRC cells (Figure 4B)”.

Beclin1 expression was significantly increased in HCR116/ST and SW620/OR compared to each parental cell (Figure 4C).

5. Page 9, lines 329-330: “The protein expression levels of PKM2 and LDHA remained comparable among the PT, ST and OR CRC cells”.

PKM2 expression was significantly decreased in HT29/ST but increased in SW480/OR compared to each parental cell (Figure 6B).

Author Response

Response to Reviewer 1

The revised paper by Park et al. has been well improved. However, some results are not precisely described based on the data. So, the description of the paper should be revised according to the data results. Several points to be clarified are below.

 Minor comments

  1. Page 5, lines 201-202: IC50 values of TSA were significantly up-regulated in H129/OR and SW620/OR compared with those of each parental cell (Table 1).

Response

Although IC50 of TSA significantly increased by 1.32- and 1.58-fold in HT29/OR and SW620/OR CRC cells, respectively, the IC50increase was not large compared to other drugs, so it seems somewhat unreasonable to conclude that resistance to TSA has developed in HT29/OR and SW620/OR CRC cells. Therefore, “Resistance to TSA was not observed in any CRC cell line (Table 1)” is revised to “The IC50 values of TSA significantly increased by 1.32- and 1.58-fold in HT29/OR and SW620/OR CRC cells, respectively, but the difference was not large compared to other drugs, so the resistance to TSA did not seem to develop in any CRC cells (Table 1)”. (Page 5, line 200-204)

  1. Page 5, lines 218-219: “The expression of BCRP was increased in OR CRC cells except HCT116/OR CRC cells (Figure 2A and 2B)”.

Response

As the reviewer pointed out, the sentence was revised to “The expression of BCRP was significantly increased in SW480/OR and SW620/OR CRC cells (Figures 2A and 2B)”. (Page 5, line 220-221)

  1. Page 7, line 256: “No changes in CDK4 expression were observed in any of the four CRC cell lines”.

Response

As the reviewer pointed out, the sentence was revised to “The expression of CDK4 was significantly increased in HCT116/ST, SW480/ST and SW620/ST CRC cells compared to their each PT CRC cells (Figures 3C and 3D)”. (Page 7, line 258-260)

  1. Page 7, lines 269-271: “However, the expression of Beclin-1, a factor involved in the initiation phase of autophagy [36], was comparable among the PT, ST and OR CRC cells (Figure 4B)”.

Response

As reviewer pointed out, the sentence was revised to “However, the expression of Beclin-1, a factor involved in the initiation phase of autophagy [36], was comparable among the PT, ST and OR CRC cells except HCT116/OR CRC cells (Figures 4B and 4C))”. (Page 7, line 273-275)

  1. Page 9, lines 329-330: “The protein expression levels of PKM2 and LDHA remained comparable among the PT, ST and OR CRC cells”.

Response

As reviewer pointed out, the sentence was revised to “The expression of PKM2 was significantly decreased in HT29/ST but increased in SW480/OR CRC cells compared to their each PT CRC cells (Figures 6A and 6B). The protein expression of  LDHA remained comparable among the PT, ST and OR CRC cells”. (Page 9, line 333-336)

Reviewer 2 Report

Dear authors, 

Thank you for answering all my suggestions and comments and for all the effort to include the data and references. In my opinion, there are a couple of comments about Figure 7 (attached file) that it would be desirable to attend.  

Author Response

Response to Reviewer 2

 Minor comments

Thank you for answering all my suggestions and comments and for all the effort to include the data and references. In my opinion, there are a couple of comments about Figure 7 (attached file) that it would be desirable to attend. 

  1. Figure 7, symbol

Response

As the reviewer pointed out, the symbol was revised in Figure 7. (Page 9, line 338)

  1. Page 10, line 355-357: “As shown in Figure 7A, co-treatment with metformin did not show any synergistic effect on the inhibition of ST CRC cell proliferation.”  

Response

As the reviewer pointed out, the sentence was modified to “As shown in Figure 7A, co-treatment with oxaliplatin and metformin did not show any synergistic effect on the inhibition of PT CRC cell proliferation”. Based on this, the next sentence was also modified to “In contrast, a co-treatment with oxaliplatin and compound C showed significantly increased the inhibition of PT CRC cell proliferation (Figure 7A)”. (Page 10, line 361-365)
